# Atmospheric Effects of Magnetosheath Jets

Alexei V. Dmitriev [1,2,]* and Alla V. Suvorova [2]

1 Department of Space Science and Engineering, National Central University, Jhongli 32001, Taiwan
2 Skobeltsyn Institute of Nuclear Physics, Lomonosov Moscow State University, 119991 Moscow, Russia
* Correspondence: dalexav@jupiter.ss.ncu.edu.tw; Tel.: +886-03-4228374

**Abstract:** We report effects in the upper high-latitude atmosphere related to the interaction of fast magnetosheath plasma streams, so-called jets, with the dayside magnetopause. The jets were observed by THEMIS mission in the dayside magnetosphere during a quiet day on 12 July 2009. It was found that the jet interaction was accompanied by strong localized compression and penetration of suprathermal magnetosheath plasma inside the dayside magnetosphere. The compression caused prominent magnetic variations with amplitudes up to 100 nT observed by ground-based magnetic networks SuperMAG and CARISMA. The magnetic variations were also visible in the geomagnetic Dst and AE indices. The jets also resulted in intense precipitation of the suprathermal ions with energies < 10 keV and energetic electrons with energies > 30 keV observed by low-altitude NOAA/POES satellites in a wide longitudinal range. The precipitations produced enhancements of ionization with an amplitude of ~1 TECU (~30% in relative units) and intensification of the ionospheric E and F1 layers as observed in the FORMOSAT-3/COSMIC misson. The enhanced ionization in the upper atmosphere might affect radio communication and navigation in the high-latitude regions. These results also provide new insight into the contribution of magnetospheric forcing to day-to-day ionospheric variability.

**Keywords:** plasma jets; particle precipitation; upper atmosphere; ionosphere

## 1. Introduction

During the last decade, there were numerous observations of fast and dense plasma streams propagating inside the magnetosheath, so-called jets [1,2]. Magnetosheath jets are transient localized plasma structures with increased velocity and density whose total energy density is several times higher than both in the surrounding magnetosheath plasma streams and in the upstream solar wind. Being fast and dense structures, jets can propagate across the magnetosheath streamlines and interact with the magnetopause resulting in its local distortion [3]. The distortion consists in local magnetopause indentation of several $R_E$ and undulation [4,5]. The interaction of jets with the magnetopause can also result in penetration of the magnetosheath plasma into the magnetosphere [6]. It has been found that the penetration can be produced by a complex mechanism of finite Larmor radius effect and impulsive plasma penetration.

Magnetosheath plasma jets are a typical consequence of the foreshock dynamics and variations in the interplanetary magnetic field (IMF) orientation [2,3,6–10]. The IMF orientation determines the location of the foreshock, the region in front of the quasi-parallel bow shock where the IMF vector is parallel to the bow shock normal [11]. In particular, if a cone angle between the IMF vector and sun–earth line is less than 30° (a case of quasi-radial IMF), the foreshock locates in the subsolar region. Processes within the foreshock significantly modify solar wind parameters such as density, velocity, and pressure [7]. Steady subsolar foreshock results in a substantial decrease in the average total (thermal and magnetic) pressure in the magnetosheath [12]. At the same time, it is in the foreshock that the majority of magnetosheath jets are generated [13]. On the other hand, large-scale jets with a duration of >30 sec are mostly generated in the interaction of IMF discontinuities

with the bow shock in the absence of significant solar wind perturbations [10]. An abrupt change in the foreshock geometry results from rapid IMF rotation, the so-called IMF discontinuity. It was found in [14] that jets of cross-sectional diameters larger than 2 Re should hit the subsolar magnetopause about several times per hour under favorable, low IMF cone angle conditions.

Previous studies showed that magnetosheath jets can be very geoeffective structures, which transmit energy, momentum and hot plasma from the solar wind into the magnetosphere [6,14,15]. Magnetospheric and ionospheric responses to jet impacts are similar to transient events occurred under abrupt solar wind pressure changes [2]. The difference consists of spatial localization of effects. It was reported that pressure pulses could be responsible for various effects in the magnetosphere and high-latitude atmosphere, such as particle precipitation at high latitudes and dayside discrete aurora, ULF waves in the magnetosphere, including Pc1 geomagnetic pulsations of pearls kind, transient field-aligned currents and traveling convection vortices in the high-latitude ionosphere and local ionospheric convection flow [15–23].

Sharp changes in the pressure at the fronts of solar wind inhomogeneities can also cause accelerated displacements of the magnetosphere boundary and trigger the buildup of the Kelvin–Helmholtz and Rayleigh–Taylor hybrid instability on the magnetopause. As a result, the impulsive penetration of plasma from the magnetosheath into the magnetosphere and the excitation of geomagnetic pulsations in the Pc2–Pc4 range occur [24,25]. The magnetosheath plasma penetration through the dayside magnetopause was described by several mechanisms, such as magnetic reconnection [26,27], impulsive penetration [28,29], finite Larmor radius [30], and a combination of them [6].

The interaction of jets with magnetopause can also result in magnetosheath plasma penetration inside the dayside magnetosphere [6]. The amount of penetrated plasma can be compared with estimates of the total amount of plasma entering the dayside magnetosphere [26]. The penetration is followed by precipitation of the suprathermal plasma to the high-latitude ionosphere in the dayside and flank sectors [31]. Analyses of dayside auroral transient events [17] showed that the dayside aurora brightening was related to localized magnetospheric compressions driven by abrupt changes in the foreshock but not by variations in the pristine solar wind dynamic pressure. Recent comprehensive and statistical studies present observations of dayside aurora brightening related to localized magnetopause indentations [21] and caused by magnetosheath high-speed jets [32]. Additionally, the study [33] provided direct evidence that the source of precipitating particles in the dayside aurora was the magnetosheath plasma (sometimes mixed with magnetospheric plasma). Thus, these studies showed that jets can be responsible for transient dayside aurora. However, there are still no direct observations of the connection between the jets and the occurrence of auroral precipitations.

In the present paper, we report the effects produced by a strong magnetosheath jet in the atmosphere. The direct observations of magnetosheath jets are presented in Section 2. The jet-related effects in the high-latitude atmosphere are described in Section 3. The results are discussed in Section 4. Section 5 is the conclusions.

## 2. Magnetosheath Jets

In order to analyze the effects of magnetosheath jets, a quiet interval on 12 July 2009 was considered. Almost the whole day was magnetically quiet (see Figure S1 in Supplementary Materials). Figure 1 shows upstream solar wind and geomagnetic conditions observed from 20 UT to 24 UT on 12 July 2009. The solar wind and interplanetary magnetic field (IMF) were observed far upstream by the ACE satellite, located near the L1 point at a geocentric distance of about 240 Earth radii (Re). As one can see in Figure 1, the solar wind was slow and quasi-state for several hours: the solar wind velocity and density varied gradually around 355 km/s and 4 cm$^{-3}$, respectively. Variations of IMF components in GSM coordinates were very weak. IMF Bz component was small and negative (around −1.5 nT) with a short (<10 min) positive excursion around 2200 UT. The IMF Bx component was quasi-state

and negative (~−1 nT). The IMF By component rotated from negative to positive values such that several rotational discontinuities occurred from 2100 to 2210 UT. The quasi-state solar wind conditions should provide very quiet geomagnetic conditions. Indeed, the Disturbance storm time (*Dst*) index was very close to 0, and Auroral Electrojet (AE) index was below 100 nT during that time. However, at 2200–2300 UT, the geomagnetic indices Dst and AE exhibited disturbances that were not related to the variations in the solar wind plasma and IMF. In particular, the Dst and AE indices increased, respectively, by 5 nT and 50 nT within the time interval from 2241 to 2253 UT.

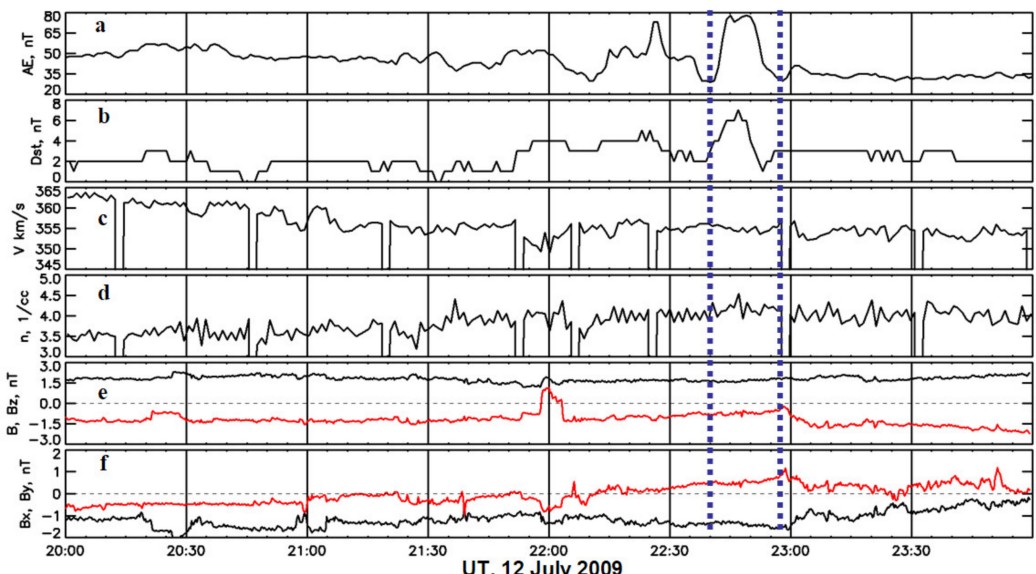

**Figure 1.** Upstream solar wind and geomagnetic conditions from 20 UT to 24 UT on 12 July 2009: (**a**) auroral electrojet (AE) index; (**b**) geomagnetic Dst index; (**c**) solar wind velocity; (**d**) solar wind density; (**e**) interplanetary magnetic field (IMF) strength (black curve) and Bz component (red curve) in GSM; (**f**) IMF components Bx (black curve) and By (red curve) in GSM. The interval of geomagnetic perturbations is indicated by the vertical blue dotted lines.

We use THEMIS data for analysis of conditions in the dayside magnetosheath and outer magnetosphere [34]. Figure 2 shows plasma and magnetic field observations conducted by the THEMIS-C probe during the geomagnetic perturbations at 2230–2315 UT on 12 July 2009. At that time, the probe was located at GSM coordinates X = 4.5 Re, Y = 14 Re and Z = −0.1 Re (the geocentric distance was 14.7 Re), i.e., it was moving in the postnoon region of the magnetosphere at local time around 1630LT. The magnetospheric region is characterized by intense fluxes (>$10^3$ (cm$^2$ s sr eV)$^{-1}$) of energetic ions (more than several keV) as shown in Figure 2d and by strong stationary magnetic field with dominant Bz component of ~20 nT as shown in Figure 2f.

At 2231–2235 UT and 2246–2251 UT, the THEMIS-C probe observed dense and fast magnetosheath plasma structures populated by suprathermal ions (several hundreds of eV) with very intense fluxes of >$10^8$ (cm$^2$ s sr eV)$^{-1}$ as shown in Figure 2d. The magnetosheath structures were characterized by large plasma velocities of $V$ > 200 km/s (see Figure 2e), and the orientation of Bz (see Figure 2f) coincided with the IMF Bz (see Figure 1e). Very important is that the structures have the ratio of total energy density $R$ > 1 (see Figure 2c). The total energy density is calculated as a sum of the kinetic, magnetic, and thermal energy density. The parameter $R$ is a ratio of the total energy density measured by THEMIS-C downstream of the bow shock to the total energy density of the upstream solar wind. Fast magnetosheath plasma structures with $R$ > 1 are defined as magnetosheath jets [3]. Jets can be generated in the interaction of the bow shock with interplanetary discontinuities [10]. In the present case, the jet can be related to discontinuities observed by ACE from 2100 to

2210 UT (see Figure 1f). Note that 60 min is the direct propagation time of the solar wind with V ~355 km/s from the ACE to the Earth bow shock (~225 Re).

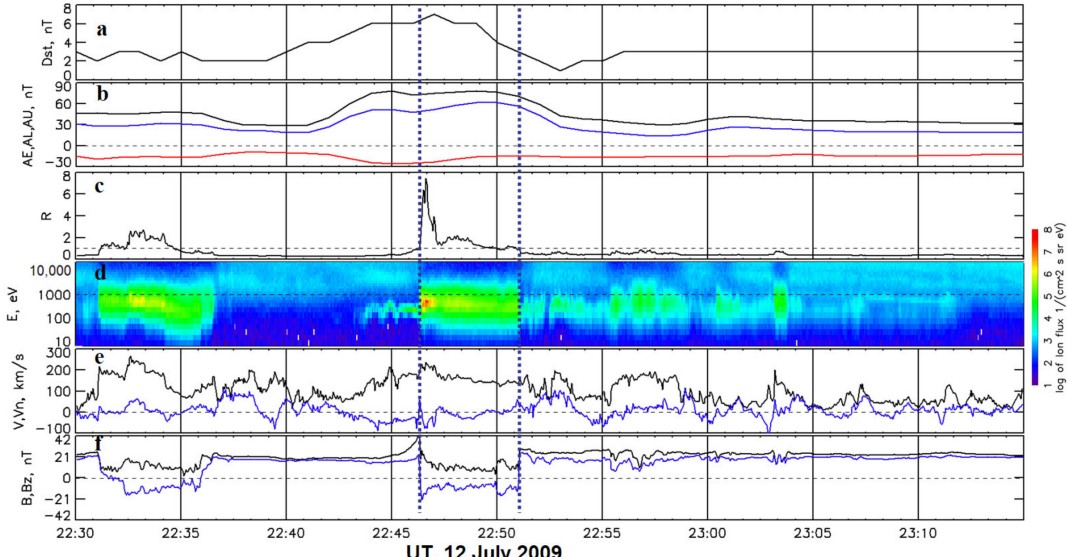

**Figure 2.** Magnetosheath plasma jets measured by THEMIS-C and geomagnetic perturbations observed from 2230 to 2300 UT on 12 July 2009: (**a**) Dst index; (**b**) auroral AE (black curve), AL (red curve) and AU (blue curve) indices; (**c**) ratio of total energy density measured downstream of the bow shock by THEMIS-C and upstream by ACE; (**d**) ion spectra measured by THEMIS-C; (**e**) THEMIS-C plasma velocity V (black curve) and normal to the magnetopause Vn (blue curve); (**f**) THEMIS-C magnetic field total B (black curve) and Bz component in GSM (blue curve). Jets were detected at 2231–2235 and 2246–2251 UT (indicated by vertical blue dotted lines).

The geoeffectiveness of jets can be estimated from the component of plasma velocity normal to the magnetopause $V$n [6]. The magnetopause surface is calculated for given solar wind conditions from a reference model [35]. The negative $V$n means that the jet moves across the magnetosheath plasma streamlines toward the magnetosphere, and, hence, the jet can hit the magnetopause. In Figure 2e, one can see that the first jet at 2231–2235 UT had a small portion with Vn ~−50 km/s and, hence, its interaction with the magnetopause was weak. The second jet at 2246–2251 UT (indicated by vertical blue dotted lines in Figure 2) was much stronger: $R$ ~7.6 and $V$n ~−100 km/s. Hence, the interaction could be strong. Note that the jet has a complex structure of the plasma streams directed inward ($V$n < 0) and outward due to deflection from the magnetopause [3,4].

The jet interaction with the magnetopause results in two effects [6]: (1) localized compression and undulation of the magnetopause and (2) direct penetration of the magnetosheath plasma inside the magnetosphere. The effect of compression can be seen as 2 time enhancement of the magnetic field strength from 21 nT to 42 nT when the jet hit the THEMIS-C probe at 2246 UT (see Figure 2f). At the same time, the Dst variation peaked at 7 nT. The enhancements of the magnetic field and Dst index were produced by strengthening of the magnetopause surface current [36], which was directly affected by the jet.

The penetration of the magnetosheath plasma inside the magnetosphere can be seen in Figure 2d. The jet was preceded (from 2244 to 2246 UT) and followed (from 2251 to 2312 UT) by intense fluxes of magnetosheath suprathermal ions with energies of hundreds of eV as observed by the THEMIS-C probe inside the magnetosphere. The appearance of magnetosheath plasma at 2244 UT, i.e., before the jet, indicates a traveling jet [3]. Namely, the jet interaction with the magnetopause started earlier in another sector. This pattern is also supported by the increase in Dst variation started at ~2240 UT, i.e., 6 min before the jet arrived at the THEMIS-C probe. According to the Dst variation, we can accept that the jet interacted with the magnetosphere and produced magnetopause undulations from 2240 to 2256 UT.

### 3. Effects in the High-Latitude Atmosphere

Variations of the auroral electroject (AE) index are shown in Figure 2b. The AE index enhanced from 2242 to 2257 UT. The enhancement is contributed by variations in both AU and AL indices, which indicates both geomagnetic compression and particle precipitation and ionization in the high-latitude atmosphere. The precipitation can be observed by low-altitude POES satellites [37]. The satellites are located at sun-synchronous orbits at altitudes ~800 km, and they measure spectra of plasma with energies from 50 eV to 10 keV and integral fluxes of energetic electrons with energy > 30 keV. The spectra are used to calculate the total energy density (TED) of suprathermal plasma fluxes. Energetic electrons are detected in two different directions: vertical and horizontal. At high latitudes, precipitating electrons propagating almost along the field lines are observed from the vertical direction.

Figure 3 shows the TED of suprathermal plasma fluxes and fluxes of energetic electrons measured at high latitudes (>60°) by five POES satellites. One can clearly see that the suprathermal plasma fluxes enhanced substantially around 2234 and, in particular, from 2243 to 2245 UT. The fluxes of energetic electrons are enhanced by two orders of magnitudes (above $10^4$ $(cm^2 s sr)^{-1}$) at 2232–2235 UT, at 2242–2245 UT, and around 2254 UT. The first two enhancements coincide in time with the jets observed by THEMIS-C. The strongest jet was accompanied by the strongest TED enhancement of ~4.5 erg/(cm² s) at ~2244 UT, and substantial enhancements of energetic electron precipitation at 2242–2245 UT were observed almost simultaneously by three POES satellites in different local time sectors: POES-18 (at ~4LT), POES-17 (at ~12LT), and POES-15 (at ~3LT).

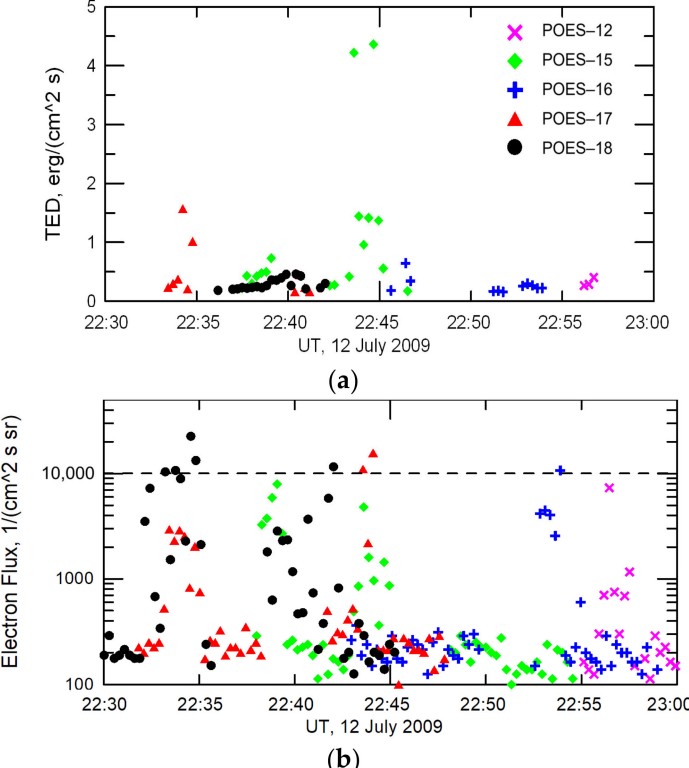

**Figure 3.** Observations of 5 NOAA/POES satellites at high latitudes (>60°) from 2230 to 2300 UT on 12 July 2009: (**a**) total energy density (TED) of the suprathermal plasma fluxes; (**b**) fluxes of >30 keV electrons.

It should be noted that the period of rotation of POES satellites is about 100 min, and they pass the high-latitude region for about 15 min at different local times. As a result, the satellites provide occasional observations of particle precipitations localized in time and space. For example, the time interval of the first jet at 2231–2235 UT was

accompanied by the enhancements observed by only two satellites, POES-17 and POES-18, located, respectively, at longitudes ~−40° (~20LT) and ~−170° (~11LT). At that time, the other satellites were located at lower latitudes. The strongest jet at 2246–2251 UT was accompanied only partially by the enhancements that occurred at 2242–2246 UT, which can be related to the earlier interaction of the jet with the magnetopause.

The spatiotemporal dynamics of magnetic disturbances at high latitudes can be investigated using data acquired from the SuperMAG ground network of magnetometers [38]. Figure 4 shows polar maps of ionospheric equivalent currents observed by the SuperMAG from 2242 to 2256 UT on 12 July 2009. The polar maps with 1-min step are shown in Figure S2 of Supplementary Materials. Before the jet impact at 2242 UT, the disturbances at high and middle latitudes were weak and did not exceed ~20 nT (see Figure 4a). At 2244 UT, the magnetic field was intensified mostly on the dayside up to amplitudes of 100 nT (see Figure 4b). Then, the region of enhanced magnetic disturbances propagated to the dusk sector, as shown in Figure 4c. Unfortunately, it is hard to analyze the disturbances in the dawn sector because of the very poor coverage of magnetic stations. The magnetic disturbances almost diminished by 2256 UT. It should be noted that the disturbances on the night side were relatively weaker during this time interval. Hence, the auroral disturbances were not produced by substorm activity.

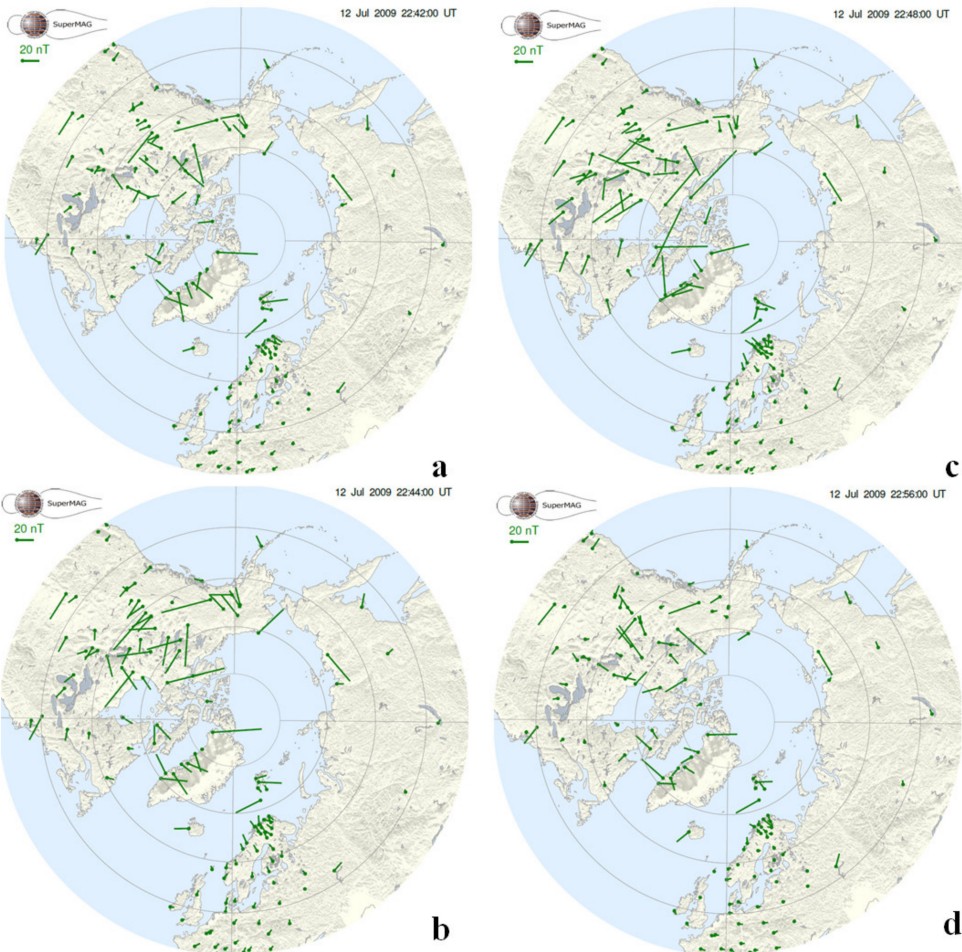

**Figure 4.** Polar maps of the equivalent ionospheric currents measured by the ground network of magnetometers SUPERMAG from 2242 to 2256 UT on 12 July 2009. The intensity of disturbances is indicated by green bars with a scale of 20 nT. The undisturbed map at 2242 UT is shown at panel (**a**). The variations were intensified at 2244 UT (panel (**b**)) on the dayside (upper hemisphere), then they propagated through flank as shown at 2248 UT (panel (**c**)) toward the nightside (lower hemisphere) and diminished at 2256 UT (panel (**d**)). The amplitude of disturbances reached up to 100 nT.

Figure 5 shows variations of the X-component of the geomagnetic field observed by three high-latitude magnetic stations situated in the postnoon sector at 21–24 UT on 12 July 2009 and widely separated in longitudes. The data were obtained from the CARISMA data repository. The stations detected strong magnetic variations with amplitudes of tens of nT peaked at about 2245UT. Table 1 lists the location of stations in geomagnetic coordinates and the peak time Tmax. One can see that Tmax increases with longitude and local time such that the 40° of the longitudinal difference between the stations ANNA and VULC takes ~2 min. Hence, the SuperMAG observations convincingly showed that during the jet interval, the magnetic disturbances at high latitudes occurred firstly in the noon region and then propagated toward flanks with the rate of 20°/min.

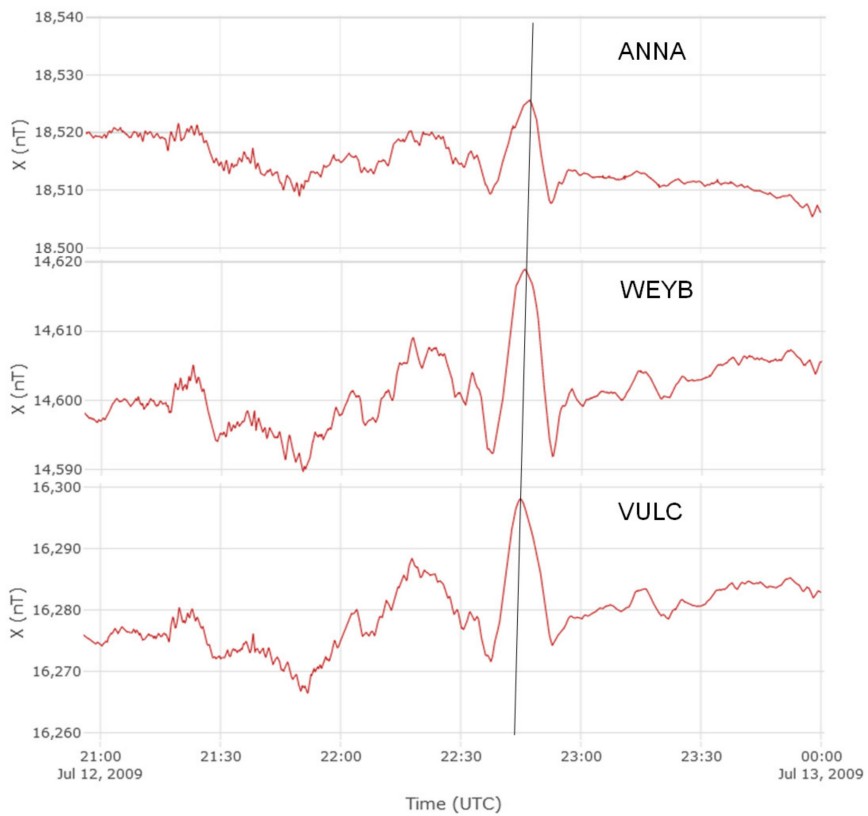

**Figure 5.** Variations of X-component of geomagnetic field observed by high-latitude magnetic stations at 21–24 UT on 12 July 2009. The peak times are connected by the black straight line.

The magnetic disturbances at high latitudes result both from the localized magnetospheric compression and from additional ionization of the upper atmosphere by precipitating particles. The spatial distributions of ionization in the upper atmosphere can be analyzed using global ionospheric maps (GIMs) of vertical total electron content (VTEC) derived from the global positioning system (GPS) network [39]. GIMs represent VTEC in geographic coordinates around the whole globe with a time step of 2 h. Figure 6 shows high-latitude regions of residual GIMs obtained on 12 July 2009. The residual GIM is calculated as a subtraction of a GIM during a quiet day from the corresponding GIM during the day of interest [40]. In the present case, the quiet day is 11 July 2009. The technique of residual GIMs allows the elimination of the background ionization produced by the solar illumination, which affects the whole high-latitude region in the Northern hemisphere during July. This makes it possible to identify the additional ionization produced in the upper atmosphere by particle precipitations and other dynamic processes in the ionosphere.

**Table 1.** Peak time of magnetic variations.

| Station | Tmax, UT | Lat, deg | Lon, deg | mLat, deg | mLon, deg | LT |
|---------|----------|----------|----------|-----------|-----------|------|
| ANNA | 22:47:16 | 42.4 | 276.1 | 52.9 | 349.5 | 1710 |
| WEYB | 22:45:47 | 49.7 | 256.2 | 58.6 | 320.9 | 1550 |
| VULC | 22:45:00 | 50.4 | 247.0 | 57.0 | 308.8 | 1514 |

　　　The background ionization at high latitudes during the undisturbed interval at 20–22 UT on 12 July 2009 is shown in Figure 6b. There are no substantial ionization enhancements except an increase of ~2 TECU in the dawn sector at longitudes from 100° to 180°. The increase is extended from the middle latitudes and, perhaps, originated from dynamical processes in the mid-latitude ionosphere. During the disturbed interval from 22 to 24 UT (Figure 6a), one can clearly see a belt of ionization enhancement of ~1 TECU observed at latitudes > 60°. The belt expands from the local noon toward dawn and dusk, where it has a maximum of >1 TECU, that is equivalent to the increase in VTEC up to 30% (see Figure S3 in Supplementary Materials). There is no additional ionization in the midnight sector. The ionization enhancement can be produced by particle precipitations during jet-related disturbances.

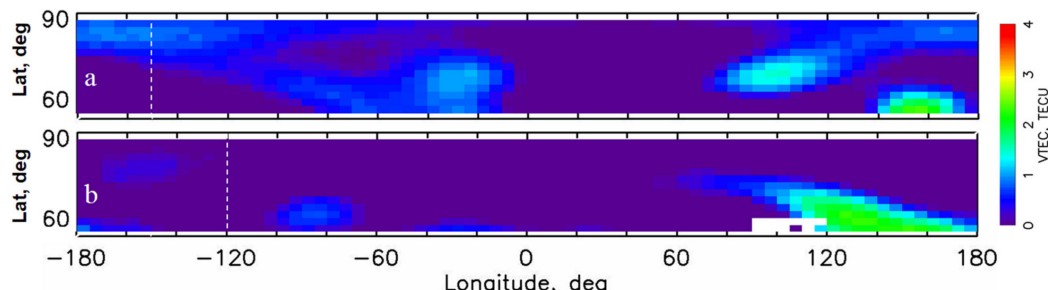

**Figure 6.** High-latitude fractions of residual global ionospheric maps (GIMs) at 20 UT (**a**) and 22 UT (**b**) on 12 July 2009. The quiet day is 11 July 2009. Vertical white dashed lines indicate local noon.

　　　The vertical distribution of the ionization can be analyzed using height profiles of electron content (EC) measured in FORMOSAT-3/COSMIC mission of six satellites located at sun-synchronous orbits [41]. EC height profiles below 800 km were measured using radio occultation (RO) of signals from the global positioning system (GPS) [42]. Figure 7 shows EC height profiles measured at high latitudes from 60° to 80° during the undisturbed time (lower panels) and disturbed time from 2243 to 2255 UT (upper panels). The total EC (TEC) is also calculated as an integral of EC along the height profile. It can be seen that in the post-midnight and pre-midnight sectors, the TEC is higher by ~1 TECU during the jet disturbed time than that during the quiet time. This number is in good agreement with the VTEC enhancements obtained from the technique of residual GIMs. The EC height profile measured at 2254 UT in the noon sector can underestimate the TEC due to a bias effect in the GPS RO technique, which results in negative values of EC [43].

　　　In Figure 7, it can be clearly seen that the additional ionization of the atmosphere occurs at altitudes below 200 km. Namely, in the noon sector (Figure 7a), the ionization increased at altitudes 150–200 km, corresponding to the F1 ionospheric region. In the post-midnight sector (Figure 7b), prominent E and F1 layers were developed at altitudes ~100 km and 150–200 km, respectively. A strong EC enhancement in the E layer was developed in the pre-midnight sector (Figure 7c) at altitudes ~100 km.

　　　Hence, the jet-related disturbances are accompanied by additional ionization in the upper atmosphere regions corresponding to the E and F1 ionospheric layers. Note that variations of the electron density in the lower ionosphere at heights below 200 km might be caused by lower atmospheric forcing such as tides, planetary waves, and gravity waves [44]. These phenomena are characterized by various temporal and spatial scales and locations. Their possible role will be considered later in Section 4.

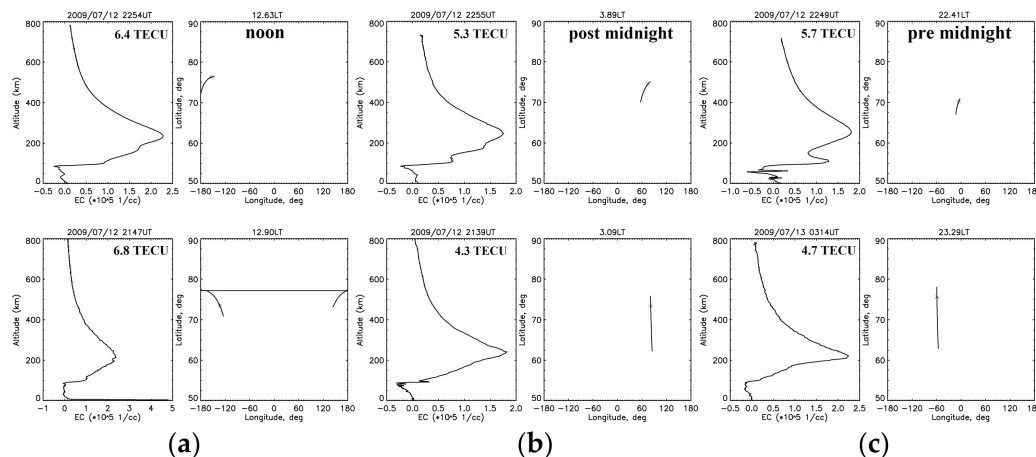

**Figure 7.** Height profiles of electron density (EC) measured by the GPS RO technique in the mission FORMOSAT-3/COSMIC at high latitudes during the jet disturbed time from 2243 to 2255 UT (upper panels) and during quiet time (lower panels) in different local time sectors: (**a**) noon; (**b**) post-midnight; (**c**) pre-midnight. Each panel shows the EC height profile (left) and the trajectory of the perigee of the RO trace (right). Asterisc indicates the EC maximum location. The total electron content is indicated for each profile in TECU.

## 4. Discussion

From the THEMIS-C observations, we found that at 2231–2235 UT and 2246–2251 UT on 12 July 2009, two magnetosheath jets interacted with the magnetopause in the postnoon sector. The jet interaction resulted in two effects in the outer magnetosphere: localized compression and penetration of magnetosheath plasma through the magnetopause. During the interaction, the following effects were observed at high latitudes: prominent magnetic disturbances, intense precipitation of suprathermal ions and energetic electrons, and additional ionization in the topside atmosphere (E and F1 regions of the ionosphere).

The high-latitude magnetic disturbances were observed simultaneously with the jet and in the same longitudinal sector. Hence, the disturbances could be originated from the jet interaction with the magnetopause. Namely, the signal from compression, which was produced by the interaction, propagated quickly in the outer magnetosphere and then with Alfven speed along magnetic field lines to the high-latitude atmosphere [45]. We found that the magnetic disturbances traveled from the noon sector toward flanks with the rate of 20°/min. This pattern can be described in the frame of the so-called traveling jet, which has the same traveling rate [3]. Figure 8 shows a sketch of the solar wind interaction with the magnetosphere during the generation of a traveling jet in the magnetosheath for the simplest case of IMF discontinuity (violet lines) transversal to the solar wind flow and moving anti-sunward with the solar wind velocity Vsw. In the present case, Vsw was ~360 km/s (see Figure 1c). At the discontinuity, the IMF By component rotates from negative to positive, as observed by ACE around 22 UT (see Figure 1f). Interaction of the discontinuity with the bow shock results in the formation of a translation region, where the foreshock translates from the prenoon to the postnoon sector. While the discontinuity is sliding along the bow shock, the translation region travels with the transversal velocity $V_t$. Apparently, $V_t$ is much higher than Vsw.

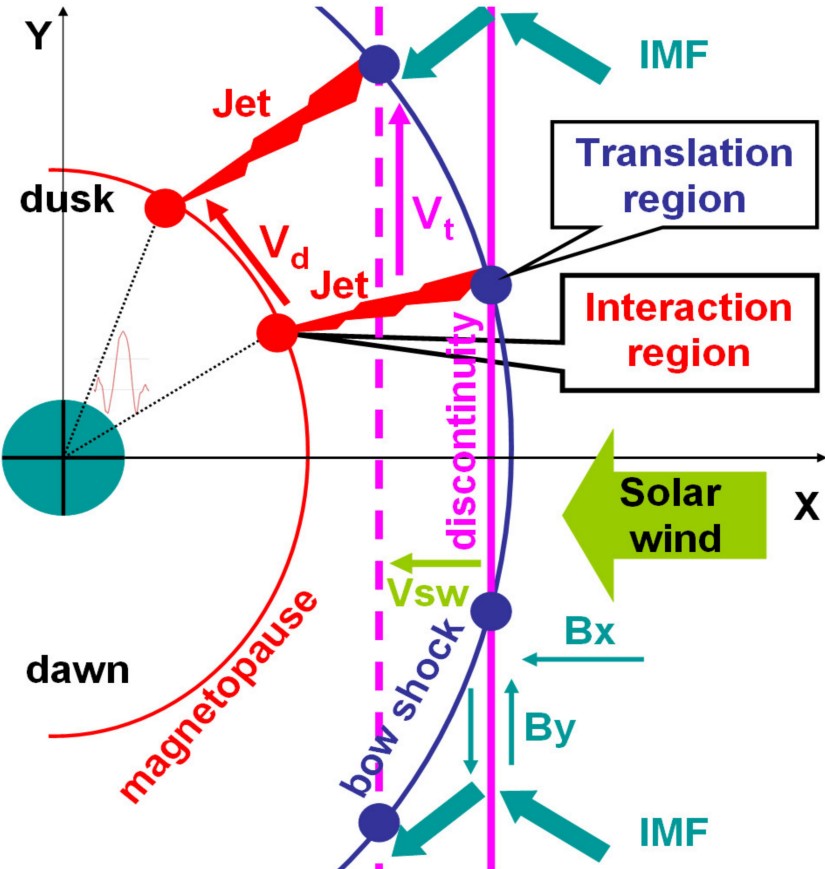

**Figure 8.** A sketch of the solar wind interaction with the magnetosphere during the generation of traveling jet in the magnetosheath. A discontinuity of the IMF By reversal from negative to positive propagates with the solar wind velocity Vsw from the noon sector (violet solid line) toward flanks (violet dashed line). Interaction of the discontinuity with the bow shock (blue curve) results in formation of a translation region (blue closed circles) where the foreshock translates from prenoon to postnoon sector. The translation region travels with the transversal velocity Vt while the discontinuity is sliding along the bow shock. A fast plasma jet (red lightning) is continuously generated in the translation region, and then it propagates across the magnetosheath streamlines and hits the magnetopause (red curve) in an interaction region (red closed circles). The interaction region, in turn, drifts along the magnetopause with velocity Vd. Disturbances in the high-latitude atmosphere of the Earth (green closed circle) are produced by the interaction region.

A fast plasma jet is continuously generated in the translation region [8]. The jet propagates across the magnetosheath streamlines and hits the magnetopause in an interaction region. The jet and the interaction region were observed by the THEMIS-C probe at geocentric distances of ~15 Re (see Figure 2). The interaction region, in turn, drifts with the jet along the magnetopause with velocity $V_d$, which is comparable with the velocity $V_t$. Assuming in the first approach the spherical dayside magnetopause, we can estimate that the interaction region should drift along the arc of 15 Re radius with the speed of 550 km/s in order to provide the traveling rate of 20°/min. This speed is much higher than the velocity of the upstream solar wind Vsw ~355 km/s (see Figure 1c), and especially the velocity of magnetosheath plasma flow (~100 km/s as shown in Figure 2e). THEMIS–C observed that the maximal speed of the plasma jet was ~240 km/s (see Figure 2e), i.e., much smaller than the traveling velocity Vd of the interaction region, which is in agreement with recent findings [46]. It was shown that jets with higher energy density (larger R) and larger spatial extension induce larger amplitude magnetic responses at high latitudes [47]. In Figure 5, one can clearly see that the first jet at 2231–2235 UT with a smaller R was accompanied by weaker magnetic variations than the strong jet at 2246–2251 UT.

The frame of the traveling jet allows an understanding of the long-lasting compression observed as positive Dst variation from 2240 to 2253 UT on 12 July 2009 (see Figure 1a). The dayside magnetopause was compressed by the jet traveling on the dayside. The solar wind with velocity Vsw ~355 km takes ~8 min to propagate from the subsolar bow shock, located at a geocentric distance of ~15 Re to the near-tail region at X ~−10 Re. The solar wind direct propagation time of ~8 min is much smaller than the 13 min duration of the compression in the Dst index, which can be explained by the tilted front of the discontinuity. Detailed analysis of the discontinuity orientation is beyond this paper.

In the topside atmosphere, we have found that the traveling jet was accompanied by ionospheric TEC enhancements with amplitude up to 30% (see Figure S3). The enhancements occurred mainly at heights below 200 km, i.e., in the ionospheric E and F1 layers (see Figure 7) and large longitudinal extensions from post-midnight through the noon to pre-midnight sectors (see Figure 6). These prominent ionospheric disturbances occurred under very quiet solar wind and geomagnetic conditions.

Variations of ionospheric ionization during quiet time have been the subject of comprehensive investigations for decades [44]. They are often prescribed for so-called day-to-day ionospheric variability produced by lower atmospheric forcing, such as tides [48], planetary waves [49], and gravity waves [50]. The effect of tides is mainly confined to lower latitudes. The planetary waves produce relatively weak variations (≤6%) in VTEC in comparison with the present case of >30% enhancement in VTEC. The gravity waves are observed in the thermosphere as traveling ionospheric disturbances (TIDs). TIDs related to lower atmospheric forcing can be produced at low and middle latitudes by such phenomena as hurricanes, tornados, tsunamis, and vulcanoes. At high latitudes, TIDs originate from auroral sources [51]. Note that the temporal scale of TIDs is of the order of hours that is much larger than the ~15 min scale of jets. The spatial scale of TIDs of several thousand kilometers is smaller than the spatial scale of the disturbances observed in relation to the traveling jet.

Recent studies of quiet-time ionospheric variability argued that the minor geomagnetic forcing produced by weak auroral sources shall play crucial roles in the mid-latitude ionosphere TEC and thermospheric composition [44]. In that study, the auroral source was characterized by a moderate increase in the AE index up to 342 nT during a quiet day (Dst ~0 and Kp < −2). It was shown that the geomagnetic forcing can penetrate from high to middle latitudes (~40N) and result in TEC variation with an amplitude of >30%. This variation is very close to the TEC enhancement observed at high latitudes during the interval of the traveling jet though the geomagnetic forcing in our case is weaker (maximum AE ~80 nT).

The ionization enhancements in the upper atmosphere at high latitudes can be produced by the precipitation of charged particles from the outer magnetosphere and radiation belt, so-called magnetospheric forcing. Precipitating ions and electrons can penetrate to various heights depending on their energies. The suprathermal ions penetrating from the magnetosheath with energies below 10 keV ionize the atmosphere at altitudes above ~150 km (ionospheric F1 region) and produce a diffuse aurora. For the ions with energy ~10 keV in the outer magnetosphere at a distance of ~15 Re, the period of azimuthal drift is very long (days), and the period of bounce motion is ~10 min. Hence, after penetration to the magnetosphere, the magnetosheath ions can precipitate into the high-latitude atmosphere after several bounces, i.e., within 10–30 min. This estimation corresponds with the ~20 min time interval during which the penetrated plasma is diminishing in the magnetosphere, as can be seen in Figure 2d. A part of the additional ionization of the high-latitude atmosphere shown in Figure 6 and enhancements of ionization in the F1 region (Figure 7a,b) can be produced by the suprathermal magnetosheath plasma precipitating from the interaction region.

The range of altitudes above 80 km (ionospheric E regions) can be ionized by energetic electrons with energies from several keV to tens of keV [52]. Energetic electrons are trapped in the outer radiation belt with the maximum at a geocentric distance of ~4 Re

that corresponds to geomagnetic latitude of ~60°. In this region, the electrons have a very short bounce period of ~1 s, and they drift azimuthally eastward with a period of several hours. During sudden pulses of the compression in the outer magnetosphere, energetic electrons are effectively scattered into the loss cone and, thus, quickly precipitate into the atmosphere at high latitudes [53]. In Figure 3b, one can see that the jets are accompanied by intense precipitations of >30 keV electrons. Apparently, the precipitation is caused by the compression produced by the interaction of the magnetopause with the traveling jets. The precipitation of energetic electrons contributes to ionization enhancements in the E region in a wide longitudinal sector on the dayside and flanks, as shown in Figures 6 and 7 b,c. It should be noted that the E layer significantly affects the propagation of radio waves. In this sense, magnetosheath plasma jets can be considered an important factor affecting radio communication and navigation in high-latitude regions. This issue will be the subject of further investigations.

## 5. Conclusions

Interaction of magnetosheath plasma jets with the magnetopause observed by THEMIS-C satellite results in the following effects in the outer atmosphere at high latitudes:

- Prominent magnetic variations with amplitudes up to 100 nT visible in the geomagnetic Dst and AE indices as well as at networks ground-based magnetic magnetometers on the dayside;
- Magnetospheric forcing in the form of intense precipitations of the suprathermal ions of more than 4 erg/(cm$^2$ s) and energetic electrons with fluxes above $10^4$ (cm$^2$ s sr)$^{-1}$;
- Enhancements of ionization with an amplitude of ~1 TECU (~30%) and intensification of the ionospheric E and F1 regions that might affect radio communication and navigation.

**Supplementary Materials:** The following supporting information can be downloaded at: https://www.mdpi.com/article/10.3390/atmos14010045/s1, Figure S1: Upstream solar wind and geomagnetic conditions on 12 July 2009; Figure S2: Polar maps of the equivalent ionospheric currents measured by the ground network of magnetometers SUPERMAG from 2241 to 2258 UT on 12 July 2009 with 1-min step.

**Author Contributions:** Conceptualization, Methodology, A.V.D.; Software, Formal analysis, A.V.S.; Writing—Original draft preparation, A.V.D.; Writing—Review and editing, A.V.S.; Visualization, A.V.S.; Project administration, A.V.D. All authors have read and agreed to the published version of the manuscript.

**Funding:** This research was funded by the MOST grants 109-2111-M-008-005 and 110-2111-M-008-013, as well as by the Research Foundation of the National Central University.

**Data Availability Statement:** The data on magnetic field and plasma measured by the THEMIS mission and ACE monitor were provided by CDAWeb (http://cdaweb.gsfc.nasa.gov/, accessed on 21 November 2022). The Dst and AE geomagnetic indices are available from http://swdcwww.kugi.kyoto-u.ac.jp/index.html, accessed on 21 November 2022. The CARISMA magnetic data are available from http://carisma.ca/carisma-data-repository, accessed on 21 November 2022. The SuperMAG magnetic data are available from http://supermag.jhuapl.edu, accessed on 21 November 2022. The NOAA/POES data are available from https://www.ngdc.noaa.gov/stp/satellite/poes/dataaccess.html, accessed on 21 November 2022. The Global Ionosphere Maps/VTEC Data are produced by European Data Center (http://ftp.aiub.unibe.ch/CODE/, accessed on 21 November 2022). The data on height profile of ionospheric electron density from COSMIC-1/FORMOSAT-3 mission are available from https://cdaac-www.cosmic.ucar.edu/cdaac/, accessed on 21 November 2022.

**Acknowledgments:** We acknowledge NASA contract NAS5-02099 and V. Angelopoulos for the use of plasma data from the THEMIS mission. We thank K.H. Glassmeier and U. Auster for the use of magnetic FGM data provided under contract 50 OC 0302 and C.W. Carlson and J.P. McFadden for the use of ESA data. We thank N. Ness and D.J. McComas for the use of ACE solar wind data made available via the CDAWeb. The authors thank Kyoto World Data Center for Geomagnetism for providing the data on geomagnetic indices. The authors thank I.R. Mann, D.K. Milling, and the

rest of the CARISMA team for the magnetometer data. CARISMA is operated by the University of Alberta and funded by the Canadian Space Agency.

**Conflicts of Interest:** The authors declare no conflict of interest.

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
