# Peer review of "Atmospheric Effects of Magnetosheath Jets"

_atmosphere, doi:10.3390/atmos14010045_

Round 1
Reviewer 1 Report
The authors provided a fairly extensive review of the literature. However, I believe that for its completeness it is useful to add on line 61 the following small text and two references:
Sharp changes in pressure at the fronts of solar wind inhomogeneities can also cause accelerated displacements of the magnetosphere boundary and trigger the buildup of the Kelvin-Helmholtz and Rayleigh-Taylor hybrid instability on it. As a result, the impulsive penetration of plasma from the magnetosheath into the magnetosphere and the excitation of geomagnetic pulsations Pc2-Pc4 occur (Mishin, JGR., 1993; Mishin et al., 2001).
References
V.V. Mishin. Accelerated Motions of the Magnetopause as a Trigger of the Kelvin-Helmholtz Instability. JOURNAL OF GEOPHYSICAL RESEARCH, VOL. 98, NO. A12, PAGES 21,365-21,371. DOI:10.1029/93JA00417
Mishin, V.V., Parkhomov, V.A., Tabanakov, I.V., Hayashi, K.: 2001, About “inclusion” of flute instability at the magnetopause during passing of the interplanetary magnetic cloud on January, 10 and 11 1997. Geomagn. Aeron. 41, 165 (in Russian). DOI:10.1007/s11207-016-0891-4
Author Response
Reply to comments of Reviewer 1
We are grateful to Reviewer 1 for very useful comments. We have revised the manuscript in according to the Reviewer’s suggestions.
Comment:
The authors provided a fairly extensive review of the literature. However, I believe that for its completeness it is useful to add on line 61 the following small text and two references:
Sharp changes in pressure at the fronts of solar wind inhomogeneities can also cause accelerated displacements of the magnetosphere boundary and trigger the buildup of the Kelvin-Helmholtz and Rayleigh-Taylor hybrid instability on it. As a result, the impulsive penetration of plasma from the magnetosheath into the magnetosphere and the excitation of geomagnetic pulsations Pc2-Pc4 occur (Mishin, JGR., 1993; Mishin et al., 2001).
Reply:
We add the following text into Introduction Section:
“Sharp changes in the pressure at the fronts of solar wind inhomogeneities can also cause accelerated displacements of the magnetosphere boundary and trigger the buildup of the Kelvin-Helmholtz and Rayleigh-Taylor hybrid instability on the magnetopause. As a result, the impulsive penetration of plasma from the magnetosheath into the magnetosphere and the excitation of geomagnetic pulsations in Pc2 - Pc4 range occur [24, 25]. The magnetosheath plasma penetration through the dayside magnetopause was described by several mechanisms such as magnetic reconnection [26, 27], impulsive penetration [28, 29], finite Larmor radius [30] and a combination of them [6].”
References:
- Mishin, V.V. Accelerated Motions of the Magnetopause as a Trigger of the Kelvin-Helmholtz Instability. Geophys. Res., 1993, 98(A12), 21,365-21,371. https://doi.org/10.1029/93JA00417
- Mishin, V.V.; Parkhomov, V.A.; Tabanakov, I.V.; Hayashi, K. About "inclusion" of flute instability at the magnetopause during passing of the interplanetary magnetic cloud on January, 10 and 11 1997. Aeron. 2001, 41, 165 (in Russian). DOI: 10.1007/s11207-016-0891-4
- Sibeck, D.G. Plasma transfer processes at the magnetopause. Space Sci. Rev. 1999, 88, 207-283. https://doi.org/10.1023/a:1005255801425
- Zhang, H.; Zong, Q; Connor, H.; Delamere, P.; Facsko, G; Han, D.; Hasegawa, H.; Kallio, E.; Kis, A.; Le, G. Dayside transient phenomena and their impact on the magnetosphere and ionosphere, Space Science Reviews. 2022, 218:40. https://doi.org/10.1007/s11214-021-00865-0
- Lemaire, J. Impulsive penetration of filamentary plasma elements into themagnetospheres of the Earth and Jupiter, Space Sci., 1977, 25, 887-890. doi:10.1016/0032-0633(77)90042-3
- Brenning, N.; Hurtig, T.; and Raadu, M. Conditions for plasmoid penetration across abrupt magnetic barriers, Plasmas, 2005, 12, 1-10. doi: 10.1063/1.1812277
- Savin, S., et al. High kinetic energy jets in the Earth's magnetosheath: Implications for plasma dynamics and anomalous transport, JETP Lett. 2008, 87(11), 593-599. doi: 10.1134/S0021364008110015

Reviewer 2 Report
1. Line 52: "geoffective" should be revised to "geoeffective".
2. Line 103: In the caption of Figure 1, "blue curve" should be revised to "red curve".
3. Figure 1: The curves of AU and AL are redundant.
4. Line 135-137: The interaction of the second jet from 2246 to 2251 UT is only observed at ~ 2246 UT, as the Vn at ~ 2251 UT is positive and even close to 0.
Author Response
Reply to comments of Reviewer 2
We appreciate the comments and corrections of Reviewer 2. We revised the text accordingly.
Comment 1. Line 52: "geoffective" should be revised to "geoeffective".
Reply: corrected
Comment 2. Line 103: In the caption of Figure 1, "blue curve" should be revised to "red curve".
Reply: corrected
Comment 3. Figure 1: The curves of AU and AL are redundant.
Reply: There are no curves of AU and AL in Figure 1. They are presented in Figure 2b. This panel is described in the beginning of Section 3 Effects in the High-Latitude Atmosphere:
“Variations of auroral electroject (AE) index are shown in Figure 2b. The AE index enhanced from 2242 to 2257 UT. The enhancement is contributed by variations in both AU and AL indices, which indicates both geomagnetic compression and particle precipitation and ionization in the high-latitude atmosphere.”
Here we stress that the increase in AE index is produced by both the compression (AU index enhancement) and by precipitations (AL index enhancement).
Comment 3. 4. Line 135-137: The interaction of the second jet from 2246 to 2251 UT is only observed at ~ 2246 UT, as the Vn at ~ 2251 UT is positive and even close to 0.
Reply: In the revised manuscript, we discuss this issue at lines 146-147 as the following:
“Note that the jet has a complex structure of the plasma streams directed inward (Vn < 0) and outward due to deflection from the magnetopause [3,4].”

Reviewer 3 Report
This paper reports a magnetosphere event: magnetosheath jets, and their possible effect on the high-latitude ionosphere during a extremely quiet period in solar minimum. Employing THEMIS data and the ACE measurements, they first determine that a magnetosheath occurred on that day. Then with the supermage magnetometer current, Global ionosphere map, cosmic electorn profile, they desribe the high-latitude ionosphere responses to this event. I am surprised to see that even during a pretty quiet day (at least from AE), there is still much ongoing outside the Earth atmosphere. This paper is potentially important to space weather forecast. The paper is well organized and clear. It shall be published after minor revisions.
Line 16 please provide the full name of Dst and AE when you first mention them
Line 90-93 here the author shall also describe the by and bx, since they are also important.
Line 209 relatively weaker
Table 1 the geographic latitude and longitude of the stations shall be listed
Line 232-258 here the author shall not only mention the value of TECU enhancement, but also provide the relative variations compared with the reference day.
Line 271-278 here the author shall also provide a paragraph to validate that the enhancement in the E and F1 region are not due to the lower atmospheric forcing. Although it is 60-80N, the lower atmospheric forcing may still play some roles. The author shall at least to provide some proof to remove the major roles of the lower atmospheric forcing. Also, in the discussion, they shall do the same thing.
Also in the discussion, the author shall acknowledge a analogous paper by Cai et al., 2021, who argued that the minor geomagnetic forcing shall still play crucial roles in the mid-latitude ionosphere TEC and thermospheric composition. The condition in Cai et al., 2021 is with AE maximum 260 nT, and the magnetospheric forcing can penetrate into mid-latitude (40N). While here AE max is only 50 nT, and the forcing just impact 60-80N. It is worth mentioning this previous study and compare with them, and to further illustrate the importance of the author's own paper here.
, , , , , , et al. (2021). Variations in thermosphere composition and ionosphere total electron content under “geomagnetically quiet” conditions at solar-minimum. Geophysical Research Letters, 48, e2021GL093300. https://doi.org/10.1029/2021GL093300
Author Response
Reply to comments of Reviewer 3
We are very grateful to Reviewer 3 for very useful comments and suggestions. They help us to improve the manuscript significantly.
Comment 1:
Line 16 please provide the full name of Dst and AE when you first mention them
Reply: The definition of abbreviations of the Dst and AE indices was done at lines 102-103:
“Indeed, the Disturbance storm time (Dst) index was very close to 0 and Auroral Electrojet (AE) index was below 100 nT during that time.”
Comment 2:
Line 90-93 here the author shall also describe the by and bx, since they are also important.
Reply: The IMF components Bx and By are described as the follows:
“The IMF Bx component was quasistate and negative (~-1 nT). The IMF By component rotated from negative to positive values such that several rotational discontinuities occurred from 2100 to 2210 UT.”
Comment 3:
Line 209 relatively weaker
Reply: Corrected
Comment 4:
Table 1 the geographic latitude and longitude of the stations shall be listed
Reply: Table 1 was revised accordingly.
Comment 5:
Line 232-258 here the author shall not only mention the value of TECU enhancement, but also provide the relative variations compared with the reference day.
Reply: We revised the paper accordingly.
- In supplementary materials, we add Figure S3, which shows relative variations in VTEC.
- The sentence was revised as the following:
“The belt expands from the local noon toward dawn and dusk, where it has the maxima of >1 TECU that is equivalent to the increase in VTEC up to 30% (see Figure S3 in supplementary materials).”
Comment 6:
Line 271-278 here the author shall also provide a paragraph to validate that the enhancement in the E and F1 region are not due to the lower atmospheric forcing. Although it is 60-80N, the lower atmospheric forcing may still play some roles. The author shall at least to provide some proof to remove the major roles of the lower atmospheric forcing. Also, in the discussion, they shall do the same thing.
Reply: We discuss this important issue in the revised manuscript.
In the end of Section 3. Effects in the High-Latitude Atmosphere, we add the following paragraph:
“Hence, the jet-related disturbances are accompanied by the additional ionization in the upper atmosphere regions corresponding to the E and F1 ionospheric layers. Note that variations of the electron density in the lower ionosphere at heights below 200 km might be caused by lower atmospheric forcing such as tides, planetary waves and gravity waves [e.g. 43]. These phenomena are characterized by various temporal and spatial scales and locations. Their possible role will be considered later in Discussion Section.”
In the Section 4. Discussion, we consider the lower atmospheric forcing in more detail:
“In the topside atmosphere, we have found that the traveling jet was accompanied by ionospheric TEC enhancements with amplitude up to 30% (see Figure S3). The enhancements occurred mainly at heights below 200 km, i.e., in the ionospheric E and F1 layers (see Figure 7) and large longitudinal extension from post-midnight through the noon to pre-midnight sectors (see Figure 6). These prominent ionospheric disturbances occurred under very quiet solar wind and geomagnetic conditions.
Variations of ionospheric ionization during quiet time is a subject of comprehensive investigations during decades [e.g. 44]. They are often prescribed to so-called day-to-day ionospheric variability produced by lower atmospheric forcing such as tides [48], planetary waves [49] and gravity waves [50]. The effect of tides is mainly confined to lower latitudes. The planetary waves produce relatively weak variations (≤6%) in VTEC in comparison with the present case of >30% enhancement in VTEC. The gravity waves are observed in the thermosphere as traveling ionospheric disturbances (TIDs). TIDs related to lower atmospheric forcing can be produced at low and middle latitudes by such phenomena as hurricanes, tornados, tsunamis and vulcanoes. At high latitudes, TIDs originate from auroral sources [51]. Note that the temporal scale of TIDs is of the order of hours that is much larger than the ~15 min scale of jets. The spatial scale of TIDs of several thousands of kilometers is smaller than the spatial scale of the disturbances observed in relation to the travelling jet.”
Comment 7:
Also in the discussion, the author shall acknowledge a analogous paper by Cai et al., 2021, who argued that the minor geomagnetic forcing shall still play crucial roles in the mid-latitude ionosphere TEC and thermospheric composition. The condition in Cai et al., 2021 is with AE maximum 260 nT, and the magnetospheric forcing can penetrate into mid-latitude (40N). While here AE max is only 50 nT, and the forcing just impact 60-80N. It is worth mentioning this previous study and compare with them, and to further illustrate the importance of the author's own paper here.
Reply: These very important results are discussed as the following:
“Recent studies of quiet-time ionospheric variability argued that the minor geomagnetic forcing produced by weak auroral sources shall play crucial roles in the mid-latitude ionosphere TEC and thermospheric composition [44]. In that study, the auroral source was characterized by a moderate increase in the AE index up to 342 nT during a quiet day (Dst ~ 0 and Kp < 2-). It was shown that the geomagnetic forcing can penetrate from high to middle latitudes (~40N) and result in TEC variation with amplitude of >30%. This variation is very close to the TEC enhancement observed at high latitudes during the interval of travelling jet though the geomagnetic forcing in our case is weaker (maximum AE ~ 80 nT).
The ionization enhancements in the upper atmosphere at high latitudes can be produced by precipitation of charged particles from the outer magnetosphere and radiation belt, so-called magnetospheric forcing…”
We also modified the conclusion:
“- magnetospheric forcing in form of intense precipitations of the suprathermal ions of more than 4 erg/(cm2 s) and energetic electrons with fluxes above 104 (cm2 s sr)-1;”
The end of abstract was also modified:
“These results also provide a new insight to the contribution of magnetospheric forcing to day-to-day ionospheric variability.”
References:
- Cai, X.; Burns, A. G.; Wang, W.; Qian, L.; Pedatella, N.; Coster, A.; et al. Variations in thermosphere composition and ionosphere total electron content under "geomagnetically quiet" conditions at solar-minimum. Res. Lett. 2021, 48, e2021GL093300. https://doi.org/10.1029/2021GL093300
- Zhang, Y.; England, S.; & Paxton, L. J. Thermospheric composition variations due to nonmigrating tides and their effect on ionosphere, Res. Lett. 2010, 37, L17103. https://doi.org/10.1029/2010GL044313
- Chang, L. C.; Yue, J.; Wang, W.; Wu, Q.; & Meier, R. R. Quasi two day wave-related variability in the background dynamics and composition of the mesosphere/thermosphere and the ionosphere. Geophys. Res.: Space Phys. 2014, 119, 4786-4804. https://doi.org/10.1002/2014JA019936
- Vadas, S. L. Horizontal and vertical propagation and dissipation of gravity waves in the thermosphere from lower atmospheric and thermospheric sources, Geophys. Res. 2007, 112. https://doi.org/10.1029/2006JA011845
- Hocke, K.; and Schlegel, K. A review of atmospheric gravity waves and travelling ionospheric disturbances: 1982-1995. Geophys. 1996, 14, 917-940. https://doi.org/10.1007/s00585-996-0917-6
